# The Effects of TNF-α Inhibition on the Metabolism of Cartilage: Relationship between KS, HA, HAPLN1 and ADAMTS4, ADAMTS5, TOS and TGF-β1 Plasma Concentrations in Patients with Juvenile Idiopathic Arthritis

**DOI:** 10.3390/jcm11072013

**Published:** 2022-04-04

**Authors:** Kornelia Kuźnik-Trocha, Katarzyna Winsz-Szczotka, Iwona Lachór-Motyka, Klaudia Dąbkowska, Magdalena Wojdas, Krystyna Olczyk, Katarzyna Komosińska-Vassev

**Affiliations:** 1Department of Clinical Chemistry and Laboratory Diagnostics, Faculty of Pharmaceutical Sciences in Sosnowiec, Medical University of Silesia, ul. Jedności 8, 41-200 Sosnowiec, Poland; kkuznik@sum.edu.pl (K.K.-T.); klaudia_092@vp.pl (K.D.); magdalena.bacia@gmail.com (M.W.); olczyk@sum.edu.pl (K.O.); kvassev@sum.edu.pl (K.K.-V.); 2Department of Rheumatology, The John Paul II Pediatric Center in Sosnowiec, ul. G. Zapolskiej 3, 41-218 Sosnowiec, Poland; iwlamo@tlen.pl

**Keywords:** juvenile idiopathic arthritis, etanercept, cartilage turnover markers, keratan sulfate, hyaluronic acid, hyaluronan and proteoglycan link protein 1, aggrecanases, transforming growth factor β1

## Abstract

We assessed the effect of 24-month anti-tumor necrosis factor alpha (TNF-α) treatment on the remodeling of the cartilage extracellular matrix (ECM) in patients with juvenile idiopathic arthritis (JIA). Methods: Quantitative evaluation of keratan sulfate (KS), hyaluronic acid (HA), hyaluronan and proteoglycan link protein 1 (HAPLN1), as potential biomarkers of joint dysfunction, and the levels of a disintegrin and metalloproteinase with thrombospondin motifs (ADAMTS) 4 and 5, total oxidative status (TOS) and transforming growth factor (TGF-β1) was performed (using immunoenzymatic methods) in blood obtained from patients before and after 24 months of etanercept (ETA) treatment. Results: When compared to the controls, KS, HA and HAPLN1 levels were significantly higher in patients with an aggressive course of JIA qualified for ETA treatment. An anti-cytokine therapy leading to clinical improvement promotes the normalization only of the HA level. Proteolytic and pro-oxidative factors, present in high concentrations in patients before the treatment, correlated with HAPLN1, but not with KS and HA levels. In these patients, negative correlations were found between the levels of TGF-β1 and KS, HA and HAPLN1. Conclusion: The anti-TNF-α therapy used in patients with JIA has a beneficial effect on ECM cartilage metabolism, but it does not completely regenerate it. The changes in the plasma HA level during the anti-cytokine therapy suggest its potential diagnostic utility in monitoring of disease activity and may be used to assess the efficacy of ETA treatment.

## 1. Introduction

Juvenile idiopathic arthritis (JIA) is an umbrella term that encompasses all forms of chronic arthritis of unknown origin and with onset in childhood. The disease constitutes the main cause of short- or long-term osteoarticular disability in children [1,2]. Among the factors involved in joint damage, proinflammatory cytokines, especially tumor necrosis factor α (TNF-α), interleukin (IL)-1β or IL-6, play a crucial role in the initiation and perpetuation of the cartilage extracellular matrix (ECM) remodeling [2].

The ECM is a multicomponent, dynamic, organized network structure with gel proper-ties whose composition and structure determine the mechanical functions and immunological properties of cartilage tissue [2,3,4,5]. Among the components of the ECM, aggrecan plays a special role in maintaining the mechanical and immunological properties of cartilage. This ability is intimately associated with the structure of aggrecan, particularly its high degree of protein core substitution with sulfated glycosaminoglycan (GAGs) chains, i.e., chondroitin sulfate (CS) and keratan sulfate (KS), as well as its ability to form large molecular aggregates in association with a chain of unsulfated hyaluronic acid (HA) [6,7].

The aggrecan core protein can be divided into discrete structural and functional globular regions, termed G1, G2 and G3, with a short, proteolytically sensitive, interglobular domain (IGD) separating regions G1 from G2. The G2 and G3 regions are separated by a long GAG-attachment region that is subdivided into three domains, i.e., two CS-rich domains (CS1 and CS2) and the KS-rich domain. The G3 globular domain is responsible for post-translational processing of the proteoglycan and its secretion, as well as for its molecular interactions with other cartilage ECM components, i.e., fibulin or tenascin. The G1 region is responsible for the interaction of aggrecan with HA, thus preventing free diffusion of the molecule within the tissue. This interaction is stabilized by the presence of a hyaluronan and proteoglycan link protein 1 (HAPLN1) that noncovalently interacts with both aggrecan (G1) and HA. The resulting aggregates have a stable macromolecular structure and provide the osmotic properties responsible for retaining water under a compressive load, consequently preventing tissue damage [6,7,8]. The aggrecan–hyaluronan aggregate can vary significantly in size depending on the length of HA and the number of available aggrecan and HAPLN1 molecules [7,9]. Loss of cartilage integrity in children with arthritis is probably associated with impaired aggrecan function due to either the proteolytic cleavage of the aggrecan core protein, which decreases aggrecan charge (CS and KS chains are released), or to the cleavage of HAPLN1 and HA, which decreases aggregate size or reduces its synthesis [6,7]. The initial stages of aggrecan degradation, resulting in disturbances in the structure and function of the articular cartilage, involve a disintegrin and metalloproteinases with thrombospondin motifs (ADAMTS) families, particularly action ADAMTS4 and ADAMTS5 [10]. ADAMTS4 and ADAMTS5 cleave aggrecan in its C-terminal CS-rich domains. However, the most detrimental cleavage is thought to occur within its IGD, generating fragments that release the entire GAGs-rich region into the synovial fluid, compromising joint function [11]. As a result, the products of the aforementioned proteolysis, i.e., GAGs, enter the bloodstream. KS concentration seems to reflect aggrecan turnover better than CS concentration. Keratan sulfates present in the blood at the highest concentration originate from cartilage structures, whereas only small amounts of KS originate from the cornea and nervous system. Chondroitin sulfates are abundant in all tissues, and their presence in blood reflects systemic changes of ECM [12,13]. The degradation of aggrecan by aggrecanases is believed to be an early, critical event in the cartilage destruction process, and precedes the probably irreversible depolymerization of the type II cartilage collagen network [14]. There is evidence that HA degradation by hyaluronidases or free radicals (FR) can also occur, particularly in the conditions of joint inflammation [15]. Hence, prompt detection of structural disorders of articular cartilage would allow clinicians to initiate appropriate therapies, which is essential for the course of the arthropathy in question. Applying the appropriate treatment too late, due to the lack of specific diagnostic biomarkers, may result in the perpetuation of pathological changes in the motor system in patients, especially in those with high disease activity who require a biological therapy. As we proved in our previous research, the biomarkers of the changes in the cartilage matrix can be both the components of the ECM and their derivatives, which are released into biological fluids in the course of metabolic transformations of these compounds [12,16,17,18,19]. We conducted studies involving children treated with methotrexate, which inhibits the development of arthropathy through a number of mechanisms, e.g., folate antagonism, adenosine signaling, generation of reactive oxygen species, decrease in adhesion molecules, alteration of cytokine and proteinases profiles or polyamine inhibition [20].

Due to the abovementioned dependencies, the aim of this research was to evaluate the dynamics of changes in the concentration of KS, HA and HAPLN1 as potential biomarkers of joint dysfunction, and the effectiveness of an anti-cytokine therapy in the blood of JIA patients both before, during and after the 24-month treatment with etanercept (ETA). The metabolism of KS, HA and HAPLN1 is related to the activity of the proteolytic enzymes from the ADAMTS group, i.e., ADAMTS-4 and ADAMTS-5, as well as the free radical activity reflected by total oxidant status (TOS) and the effects of the factor stimulating their anabolic transformations, i.e., transforming growth factor β1 (TGF-β1). Therefore, we decided to assess the interrelationships between the abovementioned variables.

## 2. Materials and Methods

### 2.1. Patients and Samples

The biological material for the research consisted of blood samples from 54 Polish Caucasian children of both sexes, aged 4–11, diagnosed with JIA and qualified for biological therapy with the use of ETA. Patients qualified for the research were treated in the Department of Rheumatology at John Paul II Pediatric Center in Sosnowiec. The JIA was diagnosed in children according to the criteria of the International League of Associations for Rheumatology [1].

Disease activity was assessed in all patients according to Juvenile Arthritis Disease Activity Score-27 (JADAS-27). The JADAS-27 (range 0–57) was calculated by summing the scores of four corset criteria: physician’s global assessment of disease activity (PGA) on a 10 cm visual analogue scale (VAS); parent/patient global assessment of well-being on a 10 cm VAS; active arthritis, defined as joint swelling or limitation of movement accompanied by pain and tenderness, assessed in 27 joints; and erythrocyte sedimentation rate (ESR). To treat the children with arthropathy, sulfasalazine (SSA, Sulfasalazin EN) at a dose of 30 mg per kilogram of body weight, prednisone (Encorton, EC) at a maximum dose of 1 mg per kilogram of body weight, and methotrexate (MTX) at a dose of ≤15 mg per square meter of body surface area were initially used, with one dose per week. The patients in whom a 3-month therapy with the use of the abovementioned drugs did not contribute to the clinical improvement were included in this study. The above evaluation was the basis for the implementation of ETA therapy in these patients (all JIA patients participated in Polish National Health Fund Therapeutic Programs employing TNF blockers—that is, B.33). Patients who qualified for ETA treatment exhibited features of one of the following forms of JIA: (1) Polyarticular JIA, in which at least 5 joints were swollen, with at least 3 joints exhibiting limited mobility and pain; increased CRP and/or ESR values; and the disease activity was assessed by a physician for at least 4 points in 10-point scale for assessing disease activity. (2) Oligoarticular JIA (extended and persistent) with poor prognosis factors, in the course of which swelling or limited mobility and pain affected at least 2 joints, and disease activity was assessed at 5 points on a 10-point scale. Other forms of JIA, as well as any other chronic and autoimmune diseases, previous treatment with biologic agents and withdrawal from the biologic therapy during the study period were considered as the exclusion criteria.

ETA was administered by subcutaneous at a dose of 0.4 m/kg body weight (up to a maximum dose of 25 mg) injection twice a week at intervals of 3–4 days, or at a dose of 0.8 mg/kg body weight (up to a maximum dose of 50 mg) once a week. In all patients, ETA was used together with the MTX, EC and SSD. After 3 months of effective therapy, both EC and SSD were withdrawn.

The assessment of circulating biomarkers of ECM alterations in the blood was performed both before the initiation of biological therapy (T0), and in the same patients, after the third (T3), the sixth (T6), the twelfth (T12), the eighteenth (T18), the twenty-fourth (T24) month of ETA treatment, i.e., after clinical improvement. The criteria for the American College of Rheumatology (ACR) improvement were used [21].

Blood samples obtained from 45 healthy individuals, with the age and sex matching the respective JIA patients, were used as controls. The healthy children who were included in our study did not suffer from any diseases which required hospitalization and did not undergo surgical procedures during the previous year. What is more, they were not treated pharmacologically just before the studies, and their results of routine laboratory tests were normal for the age group. The clinical data of healthy individuals and JIA patients enrolled in our study are shown in Table 1.

The plasma obtained both from healthy individuals and JIA patients was divided into portions and stored at −80 °C until the initiation of the study.

All subjects provided their informed consent for inclusion before they participated in the study. The study was conducted in accordance with the Declaration of Helsinki, and the protocol was approved by the Local Bioethics Committee of the Medical University of Silesia in Katowice (KNW/0022/KB/81/15).

### 2.2. The Assay of the Concentration of KS, HA, HAPLN1, ADAMTS and TGF-β1

The KS, HA, HAPLN1 ADAMTS and TGF-β1 levels were measured using blindly tested coded plasma samples in duplicate. The determination of a single parameter was completed within a day. Consequently, the inter-assay variation was insignificant.

Enzymatic immunoassays (ELISA) were used to quantify the markers of cartilage transformations following the manufacturer’s protocol. We used ELISA kits dedicated exclusively to scientific research. Plasma concentrations of KS, HAPLN1, ADAMTS4 and ADAMTS5 were determined with ELISA Kits by Cloud-Clone Corp. (Houston, TX, USA), with a minimum detection of 14.68 pg/mL (KS), 0.062 ng/mL (HAPLN1), 0.115 ng/mL (ADAMTS4) and 0.121 ng/mL (ADAMTS5). The determination of plasma HA concentration was performed with the TECO^®^Hyaluronic acid test kit, provided by TECOmedical AG (Sissach, Switzerland), with a minimum detection of 13.3 ng/mL. The plasma concentration of TGF-β1 was measured with the Human TGF-beta 1 ELISA Kit by BioVendor Research and Diagnostic Product (Brno, Czech Republic), with a minimum detection of 8.6 pg/mL. For all parameters tested, the intra-assay variability was less than 10%.

### 2.3. Statistical Analysis

Statistical analysis was performed using the Statistica 13.0 package (StatSoft, Krakow, Poland). The normality of distribution was checked by the Shapiro–Wilk test. The homogeneity of variance was assessed by Levene’s test. Variables were presented as means ± SD. A one-way analysis of variance (ANOVA) and Dunnett’s post hoc test were used to determine the significance of differences between the means in patients and controls. To compare the same parameters in each patient before and during treatment, a one-way analysis of variance (ANOVA) with repeated measures and Tukey’s post hoc test were used. Pearson’s correlation coefficient, which was modified by Bonferroni’s multivariate correction, was used for the statistical analysis of correlations between variables. *p* values < 0.05 were considered significant.

## 3. Results

The results regarding the evaluation of KS, HA, HAPLN1, ADAMTS4, ADAMTS5, TOS and TGF-β were analyzed only in JIA patients who completed the whole 24-month TNF-α therapy. The results are presented in Table 2.

### 3.1. The Plasma Levels of KS, HA and HAPLN1 in Healthy Children and JIA Patients

In patients with JIA, before anti-TNF-α therapy (T0 subgroup), we found a significant increase of the plasma concentrations of KS, HA and HAPLN1. As shown in Table 2, the untreated patients had higher levels of these markers compared to the controls, reaching 128% (*p* = 0.001), 258% (*p* = 0.007) and 60% (*p* = 0.002), respectively. What is more, it was observed that the therapy with ETA, which was employed in JIA patients, resulted in a significant decrease in plasma KS (*p* = 0.001) and HA (*p* = 0.0003). However, the concentration of KS in the blood of patients with clinically compensated disease (T24) was still significantly (*p* = 0.016) higher compared to the control. Anti-cytokine treatment led to a normalization of the concentrations of plasma HA. On the other hand, the concentration of HAPLN1 in the blood of patients after 24 months of using ETA did not differ significantly from the concentration determined before the treatment and was statistically significantly higher compared to the control (*p* = 0.020).

### 3.2. The Plasma Levels of ADAMTS-4, ADAMTS-5, TOS and TGF-β1 in Healthy Children and JIA Patients

In order to assess the factors regulating ECM metabolism, the levels of ADAMTS-4, ADAMTS-5, TOS and TGF-β1 were determined in blood samples from control and JIA patients (Table 2). The quantitative analysis of the markers showed that in the course of untreated JIA, significantly increased ADAMTS-4, ADAMTS-5, TOS and TGF-β1 levels were observed. For ADAMTS-4, ADAMTS-5, TOS and TGF-β1 levels, the mean increase was of 71% (*p* < 0.0001), 23% (*p* = 0.038), 200% (*p* < 0.0001) and 64% (*p* < 0.0001), respectively, versus the control values. It was also found that the employed ETA therapy, which resulted in suppressed inflammation, simultaneously contributed to a significant decrease (*p* < 0.05) in factors assessed based on their plasma levels compared to the pretreatment situation. The concentration of ADAMTS-4, ADAMTS-5 and TOS in the blood of treated children with JIA did not differ statistically (*p* > 0.05) from the concentration of these compounds in the group of healthy children. On the contrary, the concentration of TGF-β1 in the blood of patients after 24 months of treatment with ETA was significantly lower, about 27% (*p* = 0.005), compared to the concentration recorded in the control.

### 3.3. Correlation Analysis between Plasma KS, HA, HAPLN1 and ADAMTS4, ADAMTS5, TOS and TGF-β Levels in JIA Patients

As shown in Table 3, a correlation analysis revealed that in the untreated JIA patients (T0), there was a significant negative correlation between plasma KS level and TGF-β levels (r = −0.825, *p* = 0.012). An insignificant relationship was recorded between the GAGs plasma level and ADAMTS4 (r = 0.118, *p* = 0.700), ADAMTS5 (r = −0.205, *p* = 0.431) and TOS (r = −0.020, *p* = 0.953) concentrations, respectively. Moreover, no significant correlations were found between these variables in the same patients after anti-cytokine therapy. The values were as follows: KS with ADAMTS4 (r = −0.159, *p* = 0.651), ADAMTS5 (r = −0.143, *p* = 0.657), TOS (r = −0.225, *p* = 0.669) and TGF-β (r = −0.344, *p* = 0.403).

A correlation analysis similar to the analysis performed for KS was performed for HA (Table 3). We recorded significant relationships between HA and TGF-β levels (r = −0.813, *p* = 0.014), as well as insignificant relationships with ADAMTS4 (r = −0.139, *p* = 0.701), ADAMTS5 (r = −0.475, *p* = 0.196) and TOS (r = 0.388, *p* = 0.390) in JIA patients before ETA treatment. The relationships in the treated patients with inactive disease (T24) were as follows: HA with ADAMTS4 (r = −0.311, *p* = 0.416), ADAMTS5 (r = 0.393, *p* = 0.206), TOS (r = 0.301, *p* = 0.536) and TGF-β (r = −0.011, *p* = 0.977).

The analysis of the relations between HAPLN1 concentration and ADAMTS4, TOS and TGF- β revealed a significant relationship between these parameters in patients with untreated JIA (T0), but not in treated patients with inactive disease. In the untreated patients, the following values were observed: HAPLN1 with ADAMTS4 (r = 0.456, *p* = 0.029), TOS (r = 0.803, *p* = 0.009), and TGF-β (r = −0.761, *p* = 0.028). In this group of patients, the concentration of HAPLN1 was not significantly related to ADAMTS5 (r = 0.069, *p* = 0.740). What is more, insignificant relationships between HAPLN1 with ADAMTS4 (r = 0.365, *p* = 0.299), ADAMTS5 (r = 0.281, *p* = 0,353), TOS (r = −0.011, *p* = 0.984), and TGF-β (r = −0.4797, *p* = 0.191) were observed in the treated patients with inactive JIA (Table 3).

### 3.4. Correlation Analysis between Plasma KS, HA, HAPLN1 and JADAS-27, CRP and ESR Levels in JIA Patients

In order to achieve the main goal of the study, we assessed the relationship between the KS, HA and HAPLN1 plasma levels and the disease activity indicator, i.e., JADAS-27, as well as laboratory values of the inflammatory process indicators, i.e., CRP and ESR, in patients with JIA, both before biological treatment (T0) and after clinical improvement resulting from the applied ETA therapy (T24) (Table 3).

As a result of the statistical analysis, no significant correlation was found between plasma KS levels and the values of JADAS-27, CRP and ESR characteristic for these patients both before and after treatment. In the untreated patients, the following values were observed: KS and JADAS-27 (r = 0.285, *p* = 0.346), CRP (r = 0.208, *p* = 0.517), and ESR (r = 0.326, *p* = 0.218). The relationships in the treated patients with inactive disease were as follows: KS and JADAS-27 (r = 0.307, *p* = 0.308), CRP (r = 0.116, *p* = 0.720), and ESR (r = −0.195, *p* = 0.524). There was a significant correlation between the plasma level of HA in children with untreated JIA and the level of JADAS-27 in these patients (r = 0.738, *p* = 0.037). A similar relationship was demonstrated for CRP values (r = 0.835, *p* = 0.038) and ESR (r = 0.739, *p* = 0.015). What is more, the lack of significant correlations between the abovementioned variables was demonstrated in the group of patients after the therapy. The values were as follows: HA and JADAS-27 (r = 0.292, *p* = 0.357), CRP (r = −0.252, *p* = 0.455), and ESR (r = −0.154, *p* = 0.615). After analyzing the relationship between HAPLN1 levels and JADAS-27, CRP and ESR values, the lack of significant relationships between the abovementioned variables was demonstrated in the group of patients both before and after therapy. In the untreated patients, the following values were observed: HAPLN1 and JADAS-27 (r = 0.048, *p* = 0.927), CRP (r = 0.055, *p* = 0.872), and ESR (r = 0.094, *p* = 0.730). The relationships in the treated patients with inactive disease were as follows: HAPLN1 and JADAS-27 (r = 0.127, *p* = 0.694), CRP (r = 0.078, *p* = 0.821), and ESR (r = −0.037, *p* = 0.900).

### 3.5. Changes in Plasma Levels of KS, HA, HAPLN1, ADAMTS4, ADAMTS5, TOS and TGF-β in Patients with JIA during ETA Treatment

The levels of the assessed markers in the patients’ blood samples collected before the administration of the drug (T0) and after 3 (T3), 6 (T6), 12 (T12), 18 (T18) and 24 (T24) months of applying ETA are presented in Table 2 and visualized in Figure 1 as a percentage of the control group value.

As Table 2 indicates, changes in KS and HA levels showed a downward trend along with the duration of a biological therapy. The highest concentrations of these glycans were determined in the group of patients with active disease who qualified for biological treatment (T0), and the lowest in patients after 24 months of therapy (T24). Thus, statistically significant differences in KS plasma levels were recorded between the group of children qualified for a biological therapy (T0) and the patients from the following groups: T12 (*p* = 0.021), T18 (*p* = 0.021) and T24 (*p* = 0.001). However, in the case of HA, statistically significant differences were found between the concentration of the GAGs in the plasma of patients from the T0 group and the concentrations in patients from the following groups: T3 (*p* = 0.0005), T6 (*p* = 0.0003), T12 (*p* = 0.0002), T18 (*p* = 0.0003) and T24 (*p* = 0.0002). Significant differences in HA levels were also shown between T3 and T12 (*p* = 0.005), T3 and T18 (*p* = 0.002), T3 and T24 (*p* = 0.001), T6 and T18 (*p* = 0.011), and T6 and T24 (*p* = 0.008).

Changes in HAPLN1 levels were different in comparison to KS and HA levels. Initially, we observed a gradual increase in the concentration of this protein in the blood of afflicted children up to the sixth month of the therapy, followed by a gradual decrease. However, the differences in the determined concentrations were not statistically significant (*p* > 0.05).

Changes in the plasma levels of ADAMTS4 and ADAMTS5 observed in patients during the use of ETA allowed us to observe a slightly different sensitivity of these proteinases to an anti-cytokine therapy. Thus, the level of ADAMTS4 gradually decreased during the first 6 months of treatment, increased until the 18th month of treatment, and then decreased again after a 2-year treatment. The concentration of the assessed enzyme in the T6 group was significantly (*p* = 0.012) lower than in the T0 group. In the T24 group, the concentration of the assessed enzyme was significantly lower than in the T0 (*p* = 0.025) and T18 (*p* = 0.047) groups. The highest concentration of ADAMTS5 in T0 patients gradually decreased with the duration of treatment. Significant differences in ADAMTS5 levels were shown between T0 and T6 (*p* = 0.0005), T0 and T12 (*p* = 0.009), T0 and T18 (*p* = 0.0005), and T0 and T24 (*p* = 0.016).

Changes in TOS levels were also characterized by a downward trend during ETA treatment. The blood level of TOS in the patients from the T0 group was significantly higher than the levels characteristic of the patients from the following groups: T12 (*p* = 0.014), T18 (*p* = 0.006) and T24 (*p* = 0.0008). In the case of TGF-β, we observed a gradual decrease of the plasma level of this factor in the blood of patients during the use of ETA which was characterized by varying intensity. The highest level of TGF-β shown in the T0 group was significantly different from the levels shown in the following groups: T3 (*p* = 0.0007), T6 (*p* = 0.0002), T12 (*p* = 0.0002), T18 (*p* = 0.0002) and T24 (*p* = 0.0003).

## 4. Discussion

The progressive wear of the articular cartilage observed in the course of JIA, which contributes to the imbalance between the biological resistance of cartilage, its function and the forces acting on it, is associated with disturbances in the metabolism of the aggrecan [11]. In our earlier studies of JIA patients treated with conventional synthetic disease-modifying anti-rheumatic drugs (DMARDs), including MTX, we proved that the transformations of the mentioned proteoglycan (PG) were altered in the course of JIA [12,17]. In this study, we showed more intense quantitative changes in markers released into the bloodstream during aggrecan metabolism in patients in whom treatment with DMARDs did not result in clinical improvement. These patients were treated with a TNF-α inhibitor, i.e., ETA.

Administration of ETA for 24 months resulted in an improvement in symptoms, as evidenced by the lack of affected joints, the correct levels of CRP and ESR, and the fact that morning stiffness lasted less than 15 min. We also showed that this beneficial effect was associated with an improvement in aggrecan metabolism as assessed by the plasma levels of KS and HA in children with JIA. Indeed, a decrease in KS and HA levels was observed after anti-TNF-α treatment. However, a decrease to a normal value was recorded only for HA. The treatment had no effect on the plasma levels of HAPLN1 in children with JIA. To the best of our knowledge, this is the first study to report a relationship between a good clinical response to anti-TNF-α treatment and the plasma levels of KS, HA and HAPLN1 in JIA patients. Quantitative changes in the level of assessed GAGs in plasma, which were observed in children with JIA during a biological therapy, seem to result from the effective inhibition of the key proinflammatory cytokine, i.e., TNF-α, which initiates the degradation of cartilage in patients [22]. This factor, with a pleiotropic effect, increases the influx of leukocytes to the synovium, and stimulates the proliferation and differentiation of both T and B lymphocytes. Moreover, TNF-α, through changes in the mutual interactions between osteoclasts, osteoblasts and cells of the immune system, and by inducing the synthesis of proteinases in chondrocytes, leads to the destruction of primary articular cartilage, followed by the subchondral bone [2,22,23]. Aggrecanases from the ADAMTS family are among the proteinases involved in the degradation of the ECM components, which could result in changes in the levels of the assessed markers in the blood [11]. However, significantly high levels of both ADAMTS4 and ADAMTS5 in the blood of patients with an aggressive course of JIA qualified for ETA treatment did not correlate with the levels of KS and HA in these patients. On the other hand, a significant correlation was found between HAPLN1 and ADAMTS4 in these patients. The obtained results seem to confirm the leading role of ADAMTS4 in modeling the transformation of ECM components in pathological conditions, while ADAMTS5 seems to play an important role in the development of the osteoarticular system and the changes of aggrecan occurring in physiological conditions, which probably results from its continuous synthesis in chondrocytes and synovial fibroblasts [14,24,25].

These enzymes “cleave” a bond in the aggrecan protein core in its GAG-rich C-terminal domains. However, IGD proteolysis is the most destructive for the function of cartilage, as its products, i.e., the C-terminal fragments of core proteins with GAGs side chains attached, are released into the synovial fluid compromising joint function [14]. It was suggested that in JIA patients, aggrecanolysis occurs less efficiently in the IGD domain and more efficiently in the region rich in chondroitin sulphates [26]. As a result, the products of the aforementioned proteolysis, i.e., CS, and to a lesser extent KS, enter the bloodstream. The degradation of aggrecan by aggrecanases is an early and reversible event in contrast to the degradation of type II collagen catalyzed by metalloproteinases. Due to the abovementioned facts and the ability of aggrecan to prevent degradation of collagen type II, the inhibition of depolymerization of the discussed proteoglycan by aggrecanases may be a key therapeutic strategy to prevent the degradation of articular cartilage in children [14].

The slightly different trends in changes in the concentration of aggracanases in the blood of patients during the 2-year therapy with ETA, observed in this study, seem to confirm the possibility of variable ADAMTS4 and ADAMTS5 activity depending on their tissue localization, as well as different effects of cytokines stimulating the expression of genes of these enzymes and their activation [14,24,25,27]. IL-1α and TNF-α were found to stimulate ADAMTS5 activity mainly in the synovium and patella, but not in the cartilage of the femoral head or tibia in the joints of mice (ex vivo studies). This indicates, according to the authors, that ADAMTS5 may not be the predominant aggrecanase in articular cartilage in the course of arthritis [27]. However, in the bovine menisci, the expression of ADAMTS4 genes is preferentially increased by IL-1α in the inner zones of these cartilage and fibrous connective tissue structures, while the expression of ADAMTS5 genes is preferentially increased by TNF-α in their outer zones [28]. What is more, the expression of ADAMTS4 genes in the culture of synoviocytes from synovial tissues obtained from patients with osteoarthritis was significantly inhibited by ETA, and much more strongly inhibited when the TNF-α inhibitor was combined with a neutralizing anti-IL-1β antibody. The researchers did not find a similar effect for ADAMTS5, indicating that this enzyme, being insensitive to the effects of TNF-α and IL-1 synthesized by synovial macrophages, could be expressed in articular cartilage continuously and regardless of environmental conditions [24,29]. We confirmed the inhibitory effect of anti-cytokine therapy on ADAMTS4 and ADAMTS5 expression in JIA patients.

As a result of the increased aggrecan degradation in children with JIA stimulated by high levels of aggrecanases, particularly ADAMTS4, the supramolecular aggrecan-hyaluronan network is degraded. This hypothesis is confirmed by the relationship between ADAMTS4 and HAPLN1 in children before the use of the anti-cytokine therapy, as demonstrated in the present study. Moreover, in these patients, HAPLN1 was strongly related with TOS, which indicates a significant participation of the free radical form in the postsynetic modification of molecules in the mechanism of changing the ECM structure in the course of JIA. In patients with an aggressive course of the disease, the level of TOS was three-times higher than in healthy children. Within the cartilage tissue, free radicals stimulate chondrocyte apoptosis, which, in turn, leads to a reduction in the pool of newly synthesized ECM components, including glycans. Moreover, FRs regulate the transformation of ECM components through mechanisms related to the activation of latent forms of metalloproteinases or the inactivation of inhibitors of these proteinases. More, FRs directly contribute to the degradation of collagen, depolymerization of hyaluronic acid and disturbance of the synthesis of proteoglycan core proteins [30]. The hypothesis regarding the increased free radical activity characterizing JIA patients is confirmed by the increased levels of oxidative stress markers, i.e., the products of lipid or protein peroxidation, circulating in the blood of the patients, which has been demonstrated by other researchers [16,17,30,31,32,33].

The free radical degradation of hyaluronic acid leads to the release of low molecular weight fragments of its chain (LMW-HA) [34]. Such LMW-HA resulting from tissue HA depolymerization elicit proinflammatory responses by modulating toll-like receptor-4 or by activating NF-κB. NF-κB allows a vicious circle of chronic inflammation in JIA to occur by stimulating the expression of several proinflammatory cytokines, such as IL-1, IL-6 and TNF-α, which, in turn, induce alterations in HA metabolism [35,36,37]. Since hyaluronan is the most susceptible to degradation in the presence of FR among all GAG types, it may be assumed that the suppression of FR-mediated HA fragmentation may be the main cause of normalization of plasma HA in the course of a biological therapy, which was observed in our study, as well as in studies conducted in patients with rheumatoid arthritis [37]. Hence, HA appears to be a good indicator of articular cartilage regeneration, especially considering its strong correlation with the values of JADAS-27, CRP and ESR in ETA-untreated patients. The above relationships have not been demonstrated in the case of KS and HAPLN-1.

In addition to intensified catabolic processes, dysfunction of the ECM components in the course of JIA may be fostered by disturbed biosynthesis processes of these components stimulated by TGF-β1, i.e., the predominant form of TGF-β in articular cartilage [38,39,40]. We showed its significantly higher concentration in the blood of children qualified for the anti-cytokine therapy, which was strongly related to the levels of KS, HA and HAPLN1. Consistently high levels of active TGF-β were detected in degenerative joint conditions in adults and children with JIA [12,39,41]. Although high levels of TGF-β1 seem to reflect the processes of restoration of ECM components, this is not the case in patients with JIA. This is because the role of this pleiotropic cytokine in a healthy joint is different from that in an inflamed joint, i.e., in a pathological state, which is related to the activation of different Smad pathways. Thus, while low concentrations of TGF-β activate the Smad 2/3 pathway, high concentrations mainly stimulate the Smad 1/5/8 pathway [40]. It is known that this factor influences chondrocyte homeostasis during growth, and regulates the processes of remodeling and regeneration of cartilage in adults [38,39,40]. The discussed cytokine stimulates (by Smad 2/3 signaling) the early processes of chondrogenesis, including the condensation of chondrogens by stimulating the synthesis of fibronectin, as well as the proliferation and differentiation of chondroprogenitor cells [40,42]. It also inhibits the terminal differentiation of chondrocytes to a hypertrophic phenotype, and thus blocks calcification of the cartilage matrix, invasion of blood vessels, osteoblastic differentiation and ossification. However, TGF-β not only stimulates the synthesis of ECM components, such as type II collagen or aggrecan, but also reduces the degradation of components of cartilage ECM by increasing the activity of proteinase inhibitors or limiting the catabolic effects of IL-1 and TNF-α [38,39]. TGF-β was also shown to play a significant role in mediating chondrocyte responses to mechanical forces [40]. Thus, TGF-β-mediated anabolic signaling is a key aspect of the homeostasis of the articular cartilage matrix, especially during the period of growth. In pathological conditions, due to the altered concentrations of TGF-β to which chondrocytes are exposed, the function of TGF-β changes from a factor that blocks chondrocyte hyperplasia to a factor that facilitates the hypertrophy of the discussed cartilage cells (by Smad pathway 1/5/8) [40]. Consequently, the protective effect of TGF-β on articular cartilage is lost. TGF-β may additionally act as a stimulator of the inflammatory process as well as synovial hyperplasia and fibrosis [38]. The discussed cytokine stimulates the synthesis of proinflammatory cytokines, including IL-1 and TNF-α, by the cells of the synovial membrane [32]. Hence, the pathways of ECM transformation dependent on the discussed factor are not unidirectional and are related to its concentration [39]. The TGF-β1 blood levels in JIA patients treated with ETA, determined in this study, indicate that ETA, apart from leading to disease remission, also reduces the concentration of the assessed factor. However, ETA does not normalize the TGF-β1-dependent chondrocyte and cartilage homeostasis of ECM. Moreover, the determined TGF-β values, together with the evaluation of anti-inflammatory IL-10 in the blood of children with JIA obtained by Sznurkowska et al. [43], do not confirm the usefulness of the determination of the tested compounds in the diagnosis of arthropathy. The authors, without showing differences in the plasma concentrations of these compounds between patients and healthy people, suggested that the levels of circulating cytokines may be different from their concentrations at the site of inflammation [43].

## 5. Conclusions

To summarize, we showed, for the first time, that the anti-TNF-α therapy used in patients with JIA has a beneficial effect on ECM cartilage metabolism, but it does not completely regenerate it. The normalization of HA, but not KS and HAPLN1 plasma levels in patients, they confirm it.

What is more, the significant changes in the plasma HA, related to the JADAS27, CRP and ESR in JIA patients during the anti-cytokine therapy, suggest its potential diagnostic utility in the monitoring of disease activity, indicating that it may be used to assess the efficacy of ETA treatment. However, the lack of relationship between KS and HAPLN1 and disease activity, as expressed by the JADAS-27 index and inflammation values, does not confirm the usefulness of these ECM components as markers of cartilage destruction in children with JIA.

## Figures and Tables

**Figure 1 jcm-11-02013-f001:**
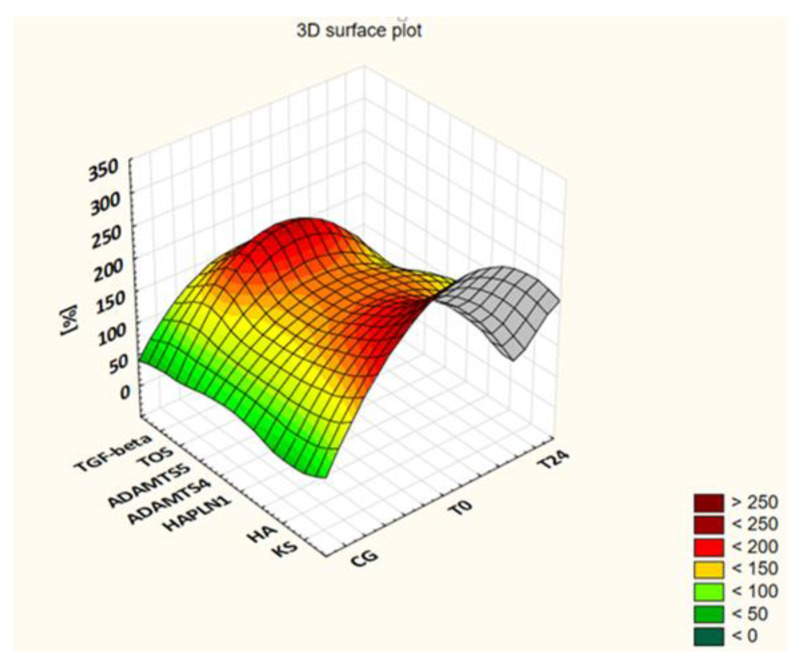
The changes in KS, HA, HAPLN1, ADAMTS4, ADAMTS5, TOS and TGF-β levels in JIA patients during ETA therapy as a percentage of the control group (CG) value.

**Table 1 jcm-11-02013-t001:** The clinical data of control subjects and JIA patients.

Parametr	Control Subjects (*n* = 45)		JIA Patients (*n* = 54)				
		Before ETA Treatment T0	Time after Starting ETA Therapy				
			3 Months T3	6 Months T6	12 Months T12	18 Months T18	24 Months T24
Age (years)	8.05 ± 2.63	6.85 ± 2.08	7.01 ± 2.10	7.35 ± 2.09	7.86 ± 2.08	8.36 ± 2.75	8.87 ± 2.10
Sex (F/M)	33/12	40/14	40/14	40/14	40/14	40/14	40/14
JADAS-27	-	41.50 (36.50–49.50)	17.50 (15.50–21.50)	9.50 (8.00–13.50)	2.50 (1.00–4.00)	1.00 (1.00–1.50)	0.50 (0.00–1.00) ^b^
Treatment drugs	-	MTX, EC, SSD	ETA, MTX, EC, SSD	ETA, MTX	ETA, MTX	ETA, MTX	ETA, MTX
WBC (10^3^/μL)	5.25 ± 2.18	9.92 ± 3.75 ^a^	7.09 ± 2.65	6.98 ± 2.88	6.74 ± 1.50	6.55 ± 1.65	6.27 ± 2.14 ^b^
RBC (10^6^/μL)	4.88 ± 0.36	3.91 ± 0.62 ^a^	4.55 ± 0.74	4.55 ± 0.85	4.51 ± 0.35	4.64 ± 0.42	4.85 ± 0.62
Hb (g/dL)	13.86 ± 1.84	11.37 ± 2.57 ^a^	12.02 ± 1.98	12.65 ± 4.45	13.47 ± 1.76	13.07 ± 1.44	13.74 ± 1.22 ^b^
PLT (10^3^/μL)	294.42 ± 72.23	349.14 ± 55.36	362.91 ± 54.01	319.15 ± 77.51	328.21 ± 85.43	312.88 ± 79.59	336.02 ± 50.42
GPT (U/L)	19.78 ± 8.02	24.02 ± 11.14	22.21 ± 7.39	17.62 ± 11.08	21.05 ± 8.27	24.51 ± 7.66	26.01 ± 10.11 ^a^
GOT (U/L)	25.74 ± 9.05	27.07 ± 11.01	26.65 ± 7.81	22.31 ± 7.43	22.17 ± 10.41	23.19 ± 10.85	25.88 ± 12.16
Cr (mg/dL)	0.71 ± 0.44	0.66 ± 0.57	0.69 ± 0.71	0.63 ± 0.24	0.72 ± 0.26	0.85 ± 0.24	0.98 ± 0.49 ^b^
ESR (mm/h)	7.02 ± 2.23	43.24 ± 13.39 ^a^	29.44 ± 13.09	12.08 ± 7.87	9.01 ± 2.53	9.68 ± 6.64	8.90 ± 1.25 ^b^
CRP (mg/L)	0.59 (0.27–1.28)	29.97 (20.82–34.16) ^a^	14.33 (11.86–16.18)	0.77 (0.37–4.82)	0.75 (0.32–3.12)	0.42 (0.20–1.20)	0.48 (0.20–1.80) ^b^

Results are expressed as mean ± SD or medians (quartile 1–quartile 3); ^a^
*p* < 0.05 compared to control group; ^b^
*p* < 0.05 compared to untreated JIA patients; ETA, etanercept; F/M, female/male; JADAS-27, Juvenile Arthritis Disease Activity Score-27; MTX, Methotrexate; EC, Encorton; SSA, Sulfasalazin; WBC, white blood cell; RBC, red blood cell; Hb, hemoglobin; PLT, platelet; GPT, glutamic pyruvic transferase; GOT, glutamic oxaloacetic transaminase; Cr, creatinine; ESR, erythrocyte sedimentation rate; CRP, C-reactive protein.

**Table 2 jcm-11-02013-t002:** The distribution patterns of plasma KS, HA, HAPLN1, ADAMTS4, ADAMTS5, TOS and TGF-β in the healthy individuals (control subjects) and JIA patients.

Parameter	Control Subjects (*n* = 45)	JIA Patients (*n* = 54)					
		Before ETA Treatment T0	Time after Starting ETA Therapy				
			3 Months T3	6 Months T6	12 Months T12	18 Months T18	24 Months T24
KS (ng/mL)	246.09 ± 35.88	561.28 ± 260.07 ^a^	493.05 ± 160.40	485.09 ± 104.63	456.27 ± 78.44 ^b^	446.38 ± 105.82 ^b^	380.60 ± 170.57 ^a,b^
HA (ng/mL)	41.76 ± 13.52	149.38 ± 94.56 ^a^	53.29 ± 24.04 ^b^	47.80 ± 23.87 ^b^	33.34 ± 15.93 ^b,c^	35.14 ± 15.85 ^b,c,d^	30.26 ± 13.25 ^b,c,d^
HAPLN1 (ng/mL)	2.35 ± 0.96	3.75 ± 1.21 ^a^	4.29 ± 1.46	4.57 ± 1.48	4.21 ± 1.29	3.75 ± 1.13	3.45 ± 0.85 ^a^
ADAMTS4 (ng/mL)	21.02 ± 9.01	38.020 ± 10.94 ^a^	32.16 ± 9.87	29.58 ± 8.98 ^b^	34.36 ± 9.55	36.68 ± 12.98	26.96 ± 10.15 ^b,e^
ADAMTS5 (ng/mL)	28.66 ± 6.66	38.09 ± 15.60 ^a^	30.08 ± 12.25	23.89 ± 7.95 ^b^	23.35 ± 8.50 ^b^	22.84 ± 6.09 ^b^	24.05 ± 5.66 ^b^
TOS (mmol/L)	439.03 ± 133.12	1316.32 ± 504.84 ^a^	714.88 ± 364.11	730.05 ± 312.28	591.08 ± 260.34 ^b^	531.88 ± 289.07 ^b^	381.18 ± 172.41 ^b^
TGF-β (ng/mL)	7.02 ± 0.95	11.48 ± 2.45 ^a^	5.67 ± 2.21 ^b^	3.94 ± 1.35 ^b^	4.32 ± 1.23 ^b^	4.51 ± 1.61 ^b^	5.11 ± 1.23 ^a,b^

Data are expressed as mean ± standard deviation; ETA, etanercept; KS, keratan sulfate; HA, hyaluronic acid; HAPLN1, hyaluronan and proteoglycan link protein 1; ADAMTS4, disintegrin and metalloproteinase with thrombospondin motifs 4; ADAMTS5, disintegrin and metalloproteinase with thrombospondin motifs 5; TOS, total oxidant status; TGF-β, transforming growth factor β; ^a^
*p* < 0.05 compared to control group; ^b^
*p* < 0.05 compared to T0 group; ^c^
*p* < 0.05 compared to T3 group; ^d^
*p* < 0.05 compared to T6 group; ^e^
*p* < 0.05 compared to T18 group.

**Table 3 jcm-11-02013-t003:** Correlation analysis between plasma KS, HA, HAPLN1 and ADAMTS4, ADAMTS5, TOS, TGF-β, JADAS-27, CRP, and ESR levels in JIA patients before ETA treatment (T0) and 24 months after starting ETA therapy (T24).

Parameter		KS	HA	HAPLN1
**JIA’ Patients before ETA Treatment (T0)**				
DAMTS4	r (*p*)	0.118 (NS)	−0.139 (NS)	0.456 (0.029)
ADAMTS5	r (*p*)	−0.205 (NS)	−0.475 (NS)	0.070 (NS)
TOS	r (*p*)	−0.020 (NS)	0.388 (NS)	0.803 (0.009)
TGF-β	r (*p*)	−0.825 (0.012)	−0.813 (0.014)	−0.761 (0.028)
JADAS-27	r (*p*)	0.285 (NS)	0.738 (0.037)	0.048 (NS)
CRP	r (*p*)	0.208 (NS)	0.835 (0.038)	0.055 (NS)
ESR	r (*p*)	0.326 (NS)	0.749 (0.015)	0.094 (NS)
**JIA’ patients 24 months after starting ETA therapy (T24)**				
ADAMTS4	r (*p*)	−0.154 (NS)	−0.311 (NS)	0.365 (NS)
ADAMTS5	r (*p*)	−0.143 (NS)	0.393 (NS)	0.281 (NS)
TOS	r (*p*)	−0.225 (NS)	0.301 (NS)	−0.011 (NS)
TGF-β	r (*p*)	−0.344 (NS)	−0.012 (NS)	−0.480 (NS)
JADAS-27	r (*p*)	0.307 (NS)	0.292 (NS)	0.172 (NS)
CRP	r (*p*)	0.116 (NS)	−0.252 (NS)	0.078 (NS)
ESR	r (*p*)	−0.195 (NS)	−0.154 (NS)	−0.037 (NS)

Results are expressed as the Pearson’s correlation coefficients; ETA, etanercept; KS, keratan sulfate; HA, hyaluronic acid; HAPLN1, hyaluronan and proteoglycan link protein 1; ADAMTS4, disintegrin and metalloproteinase with thrombospondin motifs 4; ADAMTS5, disintegrin and metalloproteinase with thrombospondin motifs 5; TOS, total oxidant status; TGF-β, transforming growth factor β; JADAS-27, Juvenile Arthritis Disease Activity Score-27; CRP, C-reactive protein; ESR, erythrocyte sedimentation rate; NS, not statistically significant.

## Data Availability

The datasets analyzed or generated during the study are available from the authors: kkuznik@sum.edu.pl or winsz@sum.edu.pl.

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
