# Peer review of "The Effects of TNF-α Inhibition on the Metabolism of Cartilage: Relationship between KS, HA, HAPLN1 and ADAMTS4, ADAMTS5, TOS and TGF-β1 Plasma Concentrations in Patients with Juvenile Idiopathic Arthritis"

_jcm, 2022, doi:10.3390/jcm11072013_

Round 1
Reviewer 1 Report
Kornelia et al. studied about The effects of TNF-α inhibition on the metabolism of cartilage: 3 relationship between KS, HA, HAPLN1 and ADAMTS5, 4 ADAMTS4, TOS and TGF-β1 plasma concentrations in patients 5 with juvenile idiopathic arthritis. This study is interesting and well wrote.
Major comment
However, authors should determine and discuss regarding KS, HA and HAPLN1, as potential biomarkers of joint dysfunction, and the effectiveness of a TNF-α inhibition therapy in the JIA patients.
Author Response
Answer to the Reviewer's comment:
We would like to thank the Reviewer for the evaluation of our article. We are grateful for the important comments, which we fully addressed in the revised manuscript. The text of our manuscript (in each individual part) has been modified, so as to facilitate its understanding and make it acceptable for publication.
Detailed modifications are presented below:
Following the Reviewer’s recommendation, to determine the role of KS, HA and HAPLN1 as potential biomarkers of joint dysfunction and the efficacy of TNF-α inhibitory therapy in JIA patients, we investigated their association with JADAS-27, CRP and ESR. [p.7, Table 3].
We have modified the part as follows:
- "Results" [p.8, lines 264-289]
2.4. Correlation Analysis Between Plasma KS, HA, HAPLN1 and JADAS-27, CRP and ESR Levels in JIA Patients
In order to achieve the main goal of the study, we assessed the relationship between the KS, HA and HAPLN1 plasma levels and the disease activity indicator, i.e. JADAS-27 as well as laboratory values of the inflammatory process indicators, i.e. CRP and ESR, in patients with JIA, both before biological treatment (T0) and after clinical improvement resulting from the applied ETA therapy (T24) (Table 3).
As a result of the statistical analysis, no significant correlation was found between plasma KS levels and the values of JADAS-27, CRP and ESR, characteristic for these patients, both before and after treatment. In the untreated patients, the following values were observed: KS and JADAS-27 (r=0.285, p=0.346), CRP (r=0.208, p=0.517), and ESR (r=0.326, p=0.218). The relationships in the treated patients with inactive disease were as follows: KS and JADAS-27 (r=0.307, p=0.308), CRP (r=0.116, p=0.720), and ESR (r=−0.195, p=0.524). There was a significant correlation between the plasma level of HA in children with untreated JIA and the level of JADAS-27 in these patients (r=0.738, p=0.037). A similar relationship was demonstrated for CRP values (r=0.835, p=0.038) and ESR (r=0.739, p=0.015). What is more, the lack of significant correlations between the above-mentioned variables was demonstrated in the group of patients after the therapy. The values were as follows: HA and JADAS-27 (r=0.292, p=0.357), CRP (r=−0.252, p=0.455), and ESR (r=−0.154, p=0.615). After analyzing the relationship between HAPLN1 levels and JADAS-27, CRP and ESR values, the lack of significant relationships between the above-mentioned variables was demonstrated in the group of patients both before and after therapy. In the untreated patients, the following values were observed: HAPLN1 and JADAS-27 (r=0.048, p=0.927), CRP (r=0.055, p=0.872), and ESR (r=0.094, p=0.730). The relationships in the treated patients with inactive disease were as follows: HAPLN1 and JADAS-27 (r=0.127, p=0.694), CRP (r=0.078, p=0.821), and ESR (r=−0.037, p=0.900).
- "Discussion" [p.11, lines 436-439]
Hence, HA appears to be a good indicator of articular cartilage regeneration. Especially in the situation of its strong correlation with the values of JADAS-27, CRP and ESR in ETA-untreated patients. The above relationships have not been demonstrated in the case of KS and HAPLN-1. [p. 11, lines 24-28].
- "Conclusion" [p.12-13, lines 493-496]
What is more, the significant changes in the plasma HA, related to JADAS27, CRP and ESR, in JIA patients during the anti-cytokine therapy, suggest its potential diagnostic utility in monitoring of disease activity, and may be used to assess the efficacy of ETA treatment. However, the lack of relationship between KS and HAPLN1 and disease activity, as expressed by the JADAS-27 index and inflammation values, does not confirm the usefulness of these ECM components as markers of cartilage destruction in children with JIA.

Reviewer 2 Report
The effects of TNF-inhibition on cartilage metabolism in JIA patients were studied by the authors. In patients with JIA and healthy controls, they measured plasma amounts of KS, HA, HAPLN1, ADAMTS5, ADAMTS4, TOS, and TGF-1.
When compared to the controls, KS, HA, and HAPLN1 levels were significantly higher in patients with JIA. “An anti-cytokine therapy leading to clinical improvement promotes the normalization only of HA level”.
Such studies have also been conducted in adult RA patients. It is one of the studies that will help us understand the destructive feature of the disease.
Some aspects of the research must be revisited.
1-In this study, how many patients with JIA were taking MTX or sulfasalazine (SZP) in addition to ETN. Treatment drugs such as MTX and SZP should be added to table 1.
2-It is well known that MTX can cause MTX osteopathy. The authors should discuss the impact of MTX on the results obtained in this study.
3-ECM proteins are extremely complex parameters. The study did not look into whether there is a link between these and CRP or erythrocyte sedimentation rate. A correlation analysis between the acute phase reactants and these parameters will be useful.
Author Response
Answer to the Reviewer's comment:
We would like to thank the Reviewer for the evaluation of our article. We are grateful for the important comments, which we fully addressed in the revised manuscript. The text of our manuscript (in each individual part) has been modified, so as to facilitate its understanding and make it acceptable for publication.
Detailed modifications are presented below:
- Following the Reviewer’s recommendation, we have modified the part of the "Materials and Methods" section. Drugs used in JIA patients have been added to Table 1. [p.4, Table 1]. The treatment of patients is described in detail in "Materials and Methods " section, as follows:
- To treat the children with arthropathy, sulfasalazine (SSA, Sulfasalazin EN) at a dose of 30 mg per kilogram of body weight, prednisone (Encorton, EC) at a maximum dose of 1 mg per kilogram of body weight, and methotrexate (MTX) at a dose of ≤15 mg, per square meter of body surface area, in one dose per week were initially used. The patients in whom a three-month therapy with the use of the above-mentioned drugs did not contribute to the clinical improvement were included into this research. [p. 3, lines 118-123].
- ETA was administered by subcutaneous injection twice a week at intervals of 3-4 days at a dose of 0.4 m /kg body weight (up to a maximum dose of 25 mg) or 0.8 mg/kg body weight (up to a maximum dose of 50 mg) once a week. In all patients, ETA was used together with the MTX, EC and SSD. After three months of effective therapy, both EC and SSD were withdrawn. [p.3, lines 136-140].
- The influence of MTX on the metabolism of ECM components was discussed in our previous research [11,15-18]. Therefore, we did not want to repeat this information. However, according to the Reviewer's recommendation we have added the information to the “Introduction” section, as follows:
We conducted studies involving children treated with methotrexate, which inhibits the development of arthropathy through a number of mechanisms, e.g. folate antagonism, adenosine signaling, generation of reactive oxygen species, decrease in adhesion molecules, alteration of cytokine and proteinases profiles or polyamine inhibition [19]. [p.2-3, lines 90-94].
- Following the Reviewer’s recommendation, we investigated association between KS, HA, HAPLN1 and JADAS-27, CRP, ESR. [p.7, Table 3, line 260]. The JADAS-27, CRP and ESR values are presented in Table 1. [p.4, line 154].
We have modified the part as follows:
- "Results" [p.8, lines 264-289]
2.4. Correlation Analysis Between Plasma KS, HA, HAPLN1 and JADAS-27, CRP and ESR Levels in JIA Patients
In order to achieve the main goal of the study, we assessed the relationship between the KS, HA and HAPLN1 plasma levels and the disease activity indicator, i.e. JADAS-27 as well as laboratory values of the inflammatory process indicators, i.e. CRP and ESR, in patients with JIA, both before biological treatment (T0) and after clinical improvement resulting from the applied ETA therapy (T24) (Table 3).
As a result of the statistical analysis, no significant correlation was found between plasma KS levels and the values of JADAS-27, CRP and ESR, characteristic for these patients, both before and after treatment. In the untreated patients, the following values were observed: KS and JADAS-27 (r=0.285, p=0.346), CRP (r=0.208, p=0.517), and ESR (r=0.326, p=0.218). The relationships in the treated patients with inactive disease were as follows: KS and JADAS-27 (r=0.307, p=0.308), CRP (r=0.116, p=0.720), and ESR (r=−0.195, p=0.524). There was a significant correlation between the plasma level of HA in children with untreated JIA and the level of JADAS-27 in these patients (r=0.738, p=0.037). A similar relationship was demonstrated for CRP values (r=0.835, p=0.038) and ESR (r=0.739, p=0.015). What is more, the lack of significant correlations between the above-mentioned variables was demonstrated in the group of patients after the therapy. The values were as follows: HA and JADAS-27 (r=0.292, p=0.357), CRP (r=−0.252, p=0.455), and ESR (r=−0.154, p=0.615). After analyzing the relationship between HAPLN1 levels and JADAS-27, CRP and ESR values, the lack of significant relationships between the above-mentioned variables was demonstrated in the group of patients both before and after therapy. In the untreated patients, the following values were observed: HAPLN1 and JADAS-27 (r=0.048, p=0.927), CRP (r=0.055, p=0.872), and ESR (r=0.094, p=0.730). The relationships in the treated patients with inactive disease were as follows: HAPLN1 and JADAS-27 (r=0.127, p=0.694), CRP (r=0.078, p=0.821), and ESR (r=−0.037, p=0.900).
- "Discussion" [p.11, lines 436-439]
Hence, HA appears to be a good indicator of articular cartilage regeneration. Especially in the situation of its strong correlation with the values of JADAS-27, CRP and ESR in ETA-untreated patients. The above relationships have not been demonstrated in the case of KS and HAPLN-1. [p. 11, lines 24-28].
- "Conclusion" [p.12-13, lines 493-496]
What is more, the significant changes in the plasma HA, related to JADAS27, CRP and ESR, in JIA patients during the anti-cytokine therapy, suggest its potential diagnostic utility in monitoring of disease activity, and may be used to assess the efficacy of ETA treatment. However, the lack of relationship between KS and HAPLN1 and disease activity, as expressed by the JADAS-27 index and inflammation values, does not confirm the usefulness of these ECM components as markers of cartilage destruction in children with JIA.

Reviewer 3 Report
Dear Author,
well explained methodology, clearly written sentences in English and I have no further suggestions and request.
Author Response
Answer to the Reviewer's comment:
We would like to thank you very much for the evaluation of our paper.

Reviewer 4 Report
Dear Editors and Authors!
Thank you for the opportunity for review the manuscript. The idea of the study is interesting, but manuscript required to be extensively elaborated and english editing required too.
The introduction should be concise and shorter, it looks like as a part of review.
The Methods and Results: please focus on the JIA characteristics (number of active joints, JIA activity indexes, e.g. JADAS, VAS, number of swollen joints, achievement of the remission, ESR, CRP, JIA categories).
The data should be presented in Median and IQR and non-parametric statistics applied.
Please provide detailed characteristics of the studied population.
I think will be better to combine the table 1 and 2, to delete the unnecessary parameters , such as RBC, Ht, RF (all negative), GPT, GOT, creatinin but add number of active joints, number of swollen joints, JADAS, the proportion of patients in the remission, to show how disease activity associated with dynamics of ECM components. Also interested the number of affected large joints (knees, ankles), presence of the erosions. According the author's theory the active large joints might be the source of ECM components in the blood, compare to small joints.
The table 3 and correlation analysis is uninformative. It much be better to compare ECM components with parameters of JIA activity (see above).
I hope the manuscript will be better after author's revisions
Author Response
Answer to the Reviewer's comment:
We would like to thank the Reviewer for the evaluation of our article. We are grateful for the important comments, which we fully addressed in the revised manuscript. The text of our manuscript (in each individual part) has been modified, so as to facilitate its understanding and make it acceptable for publication.
Detailed modifications are presented below:
- Following the Reviewer’s recommendation, we have shortened the "Introduction" section.
- Following the Reviewer’s recommendation, we have modified the "Materials and Methods " section, as follows:
- Disease activity was assessed in all patients according to Juvenile Arthritis Disease Activity Score-27 (JADAS-27). The JADAS-27 (range 0-57) was calculated by summing the scores of four core set criteria: physician’s global assessment of disease activity on a 10 cm visual analogue scale (VAS); parent/patient global assessment of well-being on a 10 cm VAS; active arthritis, defined as joint swelling or limitation of movement accompanied by pain and tenderness, assessed in 27 joints; and erythrocyte sedimentation rate (ESR). [p.3, line 112-117].
- The JADAS-27, CRP and ESR values are presented in Table 1. [p.4, line 154].
- The information about the JIA types is as follows:
Patients qualified for ETA treatment exhibited features of one of the following forms of JIA: polyarticular JIA, in which at least 5 joints were swollen, with at least 3 joints exhibiting limited mobility and pain, CRP and/or ESR values were increased, and the disease activity was assessed by a physician for at least 4 points in 10-point scale for assessing disease activity, or oligoarticular JIA (extended and persistent) with poor prognosis factors, in the course of which swelling or limited mobility and pain affect at least 2 joints, and disease activity was assessed at 5 points on a 10-point scale. Other forms of JIA as well as any other chronic and autoimmune diseases, previous treatment with biologic agents, withdrawing from a biologic therapy during the study period, were all considered as the exclusion criteria. [p.3, line 126-135].
3. We would like to inform that we presented most of the data in the form of the mean and standard deviation, as they showed a symmetrical (normal) distribution (the exception are: JADAS-25 and CRP, presented as medians and quartile 1 - quartile 3, asymmetrical distribution). Moreover, due to the size of the groups, we performed both parametric and non-parametric analyses, and obtained very similar results. Consequently, we decided to use parametric tests, and our decision can be also justified by the fact that in most cases parametric tests are more powerful.
4. Following the Reviewer’s recommendation, we have modified the characteristics of the studied population as described above (point 2). The treatment of patients is described in detail in "Materials and Methods " section, as follows:
- To treat the children with arthropathy, sulfasalazine (SSA, Sulfasalazin EN) at a dose of 30 mg per kilogram of body weight, prednisone (Encorton, EC) at a maximum dose of 1 mg per kilogram of body weight, and methotrexate (MTX) at a dose of ≤15 mg, per square meter of body surface area, in one dose per week were initially used. The patients in whom a three-month therapy with the use of the above-mentioned drugs did not contribute to the clinical improvement were included into this research. [p. 3, lines 118-123].
- ETA was administered by subcutaneous injection twice a week at intervals of 3-4 days at a dose of 0.4 m /kg body weight (up to a maximum dose of 25 mg) or 0.8 mg/kg body weight (up to a maximum dose of 50 mg) once a week. In all patients, ETA was used together with the MTX, EC and SSD. After three months of effective therapy, both EC and SSD were withdrawn. [p.3, lines 136-140].
5. We would like to inform that we did not combine Tables 1 and 2, because another Reviewer indicated the need to show changes in biochemical blood parameters during ETA treatment. However, following the Reviewer’s recommendation, we removed unnecessary parameters, such as Ht and RF from Table 1. [p.4, lines 155].
In our study, we did not differentiate patients according to the number and location of active joints. However, we found the Reviewer's suggestion very interesting and we will certainly follow it in our future research.
6. Following the Reviewer’s recommendation, we investigated association between KS, HA, HAPLN1 and JADAS-27, CRP, ESR. [p.7, Table 3].
We have modified the part as follows:
- "Results" [p.8, lines 264-289]
2.4. Correlation Analysis Between Plasma KS, HA, HAPLN1 and JADAS-27, CRP and ESR Levels in JIA Patients
In order to achieve the main goal of the study, we assessed the relationship between the KS, HA and HAPLN1 plasma levels and the disease activity indicator, i.e. JADAS-27 as well as laboratory values of the inflammatory process indicators, i.e. CRP and ESR, in patients with JIA, both before biological treatment (T0) and after clinical improvement resulting from the applied ETA therapy (T24) (Table 3).
As a result of the statistical analysis, no significant correlation was found between plasma KS levels and the values of JADAS-27, CRP and ESR, characteristic for these patients, both before and after treatment. In the untreated patients, the following values were observed: KS and JADAS-27 (r=0.285, p=0.346), CRP (r=0.208, p=0.517), and ESR (r=0.326, p=0.218). The relationships in the treated patients with inactive disease were as follows: KS and JADAS-27 (r=0.307, p=0.308), CRP (r=0.116, p=0.720), and ESR (r=−0.195, p=0.524). There was a significant correlation between the plasma level of HA in children with untreated JIA and the level of JADAS-27 in these patients (r=0.738, p=0.037). A similar relationship was demonstrated for CRP values (r=0.835, p=0.038) and ESR (r=0.739, p=0.015). What is more, the lack of significant correlations between the above-mentioned variables was demonstrated in the group of patients after the therapy. The values were as follows: HA and JADAS-27 (r=0.292, p=0.357), CRP (r=−0.252, p=0.455), and ESR (r=−0.154, p=0.615). After analyzing the relationship between HAPLN1 levels and JADAS-27, CRP and ESR values, the lack of significant relationships between the above-mentioned variables was demonstrated in the group of patients both before and after therapy. In the untreated patients, the following values were observed: HAPLN1 and JADAS-27 (r=0.048, p=0.927), CRP (r=0.055, p=0.872), and ESR (r=0.094, p=0.730). The relationships in the treated patients with inactive disease were as follows: HAPLN1 and JADAS-27 (r=0.127, p=0.694), CRP (r=0.078, p=0.821), and ESR (r=−0.037, p=0.900).
- "Discussion" [p.11, lines 436-439]
Hence, HA appears to be a good indicator of articular cartilage regeneration. Especially in the situation of its strong correlation with the values of JADAS-27, CRP and ESR in ETA-untreated patients. The above relationships have not been demonstrated in the case of KS and HAPLN-1. [p. 11, lines 24-28].
- "Conclusion" [p.12-13, lines 493-496]
What is more, the significant changes in the plasma HA, related to JADAS27, CRP and ESR, in JIA patients during the anti-cytokine therapy, suggest its potential diagnostic utility in monitoring of disease activity, and may be used to assess the efficacy of ETA treatment. However, the lack of relationship between KS and HAPLN1 and disease activity, as expressed by the JADAS-27 index and inflammation values, does not confirm the usefulness of these ECM components as markers of cartilage destruction in children with JIA.
7.An English native speaker has reviewed the manuscript and language mistakes have been corrected.

Round 2
Reviewer 4 Report
The manuscript became better and might be accepted in the revised form